# Attempt to Identify Sex Hormones in the Bodies of Selected Norway Spruce Bark Beetles

**Magdalena Kacprzyk** [1,*], **Bartłomiej Bednarz** [1] and **Maciej Choczyński** [2]

1 Department of Forest Ecosystems Protection, University of Agriculture in Krakow, Al. 29 Listopada 46, 31-425 Kraków, Poland; bartlomiej.bednarz@urk.edu.pl
2 Laboratory of Ecochemistry and Ecotoxicology, Institute of Environmental Sciences, Jagiellonian University, Gronostajowa 7, 30-387 Kraków, Poland; maciej.choczynski@uj.edu.pl
* Correspondence: magdalena.kacprzyk@urk.edu.pl

**Abstract:** A gas chromatography technique was applied to the adults of *Ips typograhus* (L.) and *Pityogenes chalcographus* (L.) collected from pheromone traps placed in Norway spruce stands in southern Poland in distinguished population swarming periods for the qualitative and quantitative determination of steroid compound differences between insect sexes. Ten not yet identified for bark beetle compounds from the group of sterols, including cholestenone, 4,6-cholestadiene-3-one, choles-4-en-3,6-dione and 17β-Hydroxyandrosta-1,4-dien-3-one benzoate, which can potentially act as gender hormones were detected. The presence of ecdysone and 20-hydroxyecdysone in the bodies of the studied bark beetles was confirmed. However, slight differences in the content of ecdysteroids in the bodies of males and females may be only the remains of the insect's metamorphosis. Due to the small differences in the extracted compounds between the females and males, their variability in concentrations during the swarming period seems to be useless as a basis for sex determination.

**Keywords:** cambioxylophages; gender determination; ecdysteroids; sterols; gas chromatography





## 1. Introduction

It is considered that insect gonads do not produce sex hormones that condition the development of secondary sex characteristics, because the gender of an insect is genetically determined [1,2]. However, insects produce androgens and estrogens [3]. In some insects, an androgenic hormone, which allows the female to develop into a male (e.g., in *Lampyris noctiluca* L. (Coleoptera: Lampyridae) species), was isolated. The same hormones appear to be used by insects in molting and growth control [4]. It is also known that egg laying is controlled by hormones and pheromones, as is the case for *Rhodnius prolixus* (Stal) (Hemiptera: Reduviidae), *Schistocerca gregaria* (Forskål) (Orthoptera: Acrididae) or *Melanoplus sanguinipes* (Fabr.) (Orthoptera: Acrididae) [5]. Insect hormones can be synthesized in different tissues, including the gonads [6]. Some substances like methyl-3-buten-2-ol, considered a principal aggregation pheromone of a few conifer bark beetles in Eurasia, are male-specific compounds, and at the same time, they are also produced by the host trees [7–11].

However, in most insect species, the sex and maturity are determined and cannot be altered by hormones. The hormones in insects have an important role in the reproduction cycle [12]. They control ovary growth and the synthesis of vitellogenin [13]. The most common hormones regulating the reproductive cycle are juvenile hormones produced by the corpora allata, as well as ecdysone originating from the ovaries [14]. So far, no detailed studies on the occurrence of sex hormones in bark beetles (Coleoptera: Scolytinae) have been undertaken. Periodic outbreaks of bark beetle pests, often occurring as a result of adverse effects on the Norway spruce stands' abiotic factors, constitute a serious ecological and economic problem in European forests [15–21]. Therefore, is important to improve the methods of forest protection, including harmful forest insect monitoring and preventing

their gradation using pheromone-baited traps. Outbreaks of bark beetles are assessed with pheromone trapping, while the sex ratio of the caught insects is evaluated as a basic element of pest management in forests.

Trapping bark beetles enables the tracking of seasonal and multi-seasonal population sizes, densities and dynamics, as well as pest suppression [22–26]. The gender structure of bark beetles is caught in pheromone traps, as the assessment of population abundance helps evaluate the phase gradation of the insect pest and is the basis for the development of stand protection methods [27–29]. Sex determination of some bark beetle species is, however, quite complicated, because a visible sexual dimorphism either does not occur, or the morphological differences are inconspicuous [30]. This situation refers, for example, to the Norway spruce bark beetle (*Ips typographus* L.), the most dangerous cambiophagous insect pest in Norway spruce stands in Europe from an economic point of view [31]. Only a few authors [32–34] suggested different structures of *I. typographus* males and females, that being the hair coverage of the front edge of the top of the head and the construction of "faces" and hirsutism of the pronotum (males have a larger hillock in the central part and a less densely hairy pronotum) [32]. Quoted slight differences in the morphological structures of male and female insects allowing the evaluation of gender, however, require some experience. Much more reliable methods, although still laborious and time-consuming, are the volume–weight index method [35] or genital extraction. The specificity of many beetle catches, like *I. typographus* and other bark beetle pest species such as *Pityogens chalcographus* (L.), to pheromone traps (tens of thousands of specimens per single trap in a year) in Norway spruce stands, however, prevents a detailed analysis of all the collected material and only typically refers to a random, more or less representative set of the sampled population of insects. Therefore, it is extremely important to improve and simplify the procedures for bark beetle sex determination. The qualitative and quantitative determination of the substances characteristic to male and female bark beetles allows for quick sex ratio assessment. So far, no studies have been conducted on the occurrence of hormones produced by female ovaries and the testicles of male bark beetles, which could be potentially useful in sex identification procedures. Information on the occurrence and role of similar chemicals in insects relates only to juvenile (neotenin) and molting (ecdysone) hormones, which are of practical use in forest protection against insect defoliators [36]. Despite the opinion on the genetic conditioning of sex in insects [37–39], the mechanism of its formation remains unresolved [40].

According to De Loof and Huybrechts [1], Raikhel et al. [41] and De Loof [4], ecdysteroids produced in the insects' reproductive organs, such as 20-hydroxyecdysone (20E), can act as sex hormones. Moreover, the role of sex hormones is represented in insects by juvenile hormones produced by the corpora allata. Juvenile hormones are in fact involved in the process of reproduction; they affect the females' sexual maturation, stimulating the growth of the ovaries before the formation of the yolk in the egg, and they are responsible for the secretion of vitellogenin [42–44]. According to Hagedorn et al. [14], Engelman [12] and Yin et al. [45], ecdysone and ecdysterone, produced by the ovaries, and other steroid hormones are important compounds involved in the reproduction process in insects. The possibility of sex differentiation in insects due to the existence of crosstalk between ecdysone signaling and the path of sex determination is discussed [46]. It is therefore highly probable that the presence of these compounds, which are responsible for the ability to reproduce, may be an important gender factor. The hypothesis of a higher concentration of 20-OH ecdysone, as well as compounds from the JH group, in the bodies of adult females preparing to lay eggs, than males has been assumed. Simultaneously, the specificity of the bark beetle swarming (i.e., the presence of few generations in a given season) and the insects' feeding period can have an direct impact on the presence and concentration of the substances differentiating sex [47]. Zumr and Soldán [48] proved that the longer the maturation feeding by females of the *I. typographus* and *P. chalcographus* species, the more eggs are present in their oviducts.

The aims of this study are (1) qualitative and quantitative determination of steroid compounds, including ecdysteroids, for sex differentiation of the most dangerous European bark beetles species; (2) determination of differences in the chemical composition of female and male bark beetles, depending on the insect swarming period; and (3) refining the chromatography procedures for the preparation and analysis of bark beetle imagines for the extraction of sex-differentiating substances in insects.

## 2. Materials and Methods

In 2013, adults of the *I. typographus* and *P. chalcographus* species, originating from pheromone traps, were placed in 100-year-old Norway spruce (*Picea abies* L. Karst) stands at an altitude of 1100 m a.s.l. in southeast Poland (Beskid Żywiecki mountains, Western Carpathians) (49° 54′ 66.17″ N, 19° 29′ 80.03″ E) and analyzed. Flight barrier Theysohn® pheromone traps were used for all species, baited with pheromone lures—Ipsodor® (Chemipan, Warsaw) for *I. typographus* and Chalcodor® (Chemipan, Warsaw) for *P. chalcographus*—and replaced once during the insects' catch inspection period on July 15. In total, for each species and localization, three pheromone traps were set. Each of the traps was checked every seven days, and the alive trapped beetles were collected. For the captured beetles in the period from May 18 to September 1, the standard procedures used in chromatography analyses were followed, being kept at a constant temperature (−25 °C) and humidity conditions (5%) in the freezer until they were divided into sexes, counted and analyzed for the presence of hormonal substances. Based on the dynamics of the insects caught, three swarming periods were established, corresponding to the generations: the first (from 16 May to 30 June), the sister (from 7 July to 21 July) and the second (from 28 July to 1 September) (Figure 1). For further analysis, random insects were chosen from each swarming period until a target sample weight of 10 g for each of the sexes was obtained (total of 60 g). The number of beetles of a given species and sex for the chromatographic analyses was 1 g, which corresponded to about 150 specimens of *I. typographus* and 2000 specimens of *P. chalcographus* [35,49]. In total, 36 samples for each insect species were analyzed. Gender identification of the *P. chalcographus* adults was based on the size of the teeth on the elytral humerus, which is bigger in males in contrast to the much smaller ones in females [50]. In the case of the *I. typographus* specimens, assignment of sex was based on genital organ extraction.

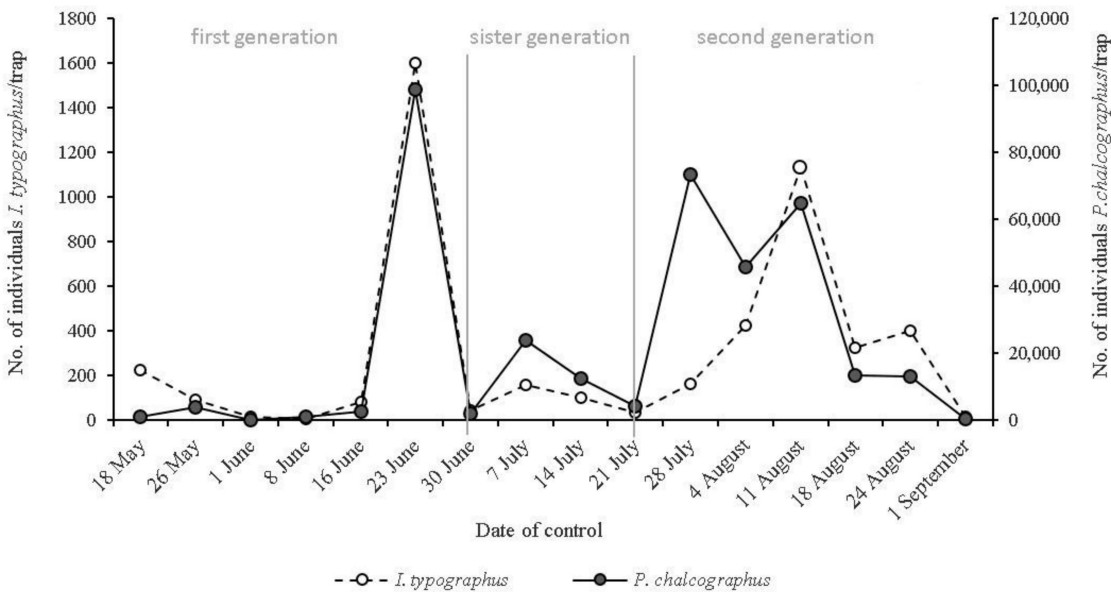

**Figure 1.** Phenology and voltinism of *I. typographus* and *P. chalcographus* in 2013, based on weekly mean captures by pheromone-baited traps (left axis: *I. typ.*; right axis: *P. chalc*). June (first peak) shows the emergence of overwintering parents and swarming of the first generation. June (second peak) shows the reemergence of parents and swarming of sister generations. July (third peak) shows the swarming of the second generation.

The studies looked for compounds differing sex, which could act as hormones (especially sex). This group includes compounds from the group of cholesterol-derived steroids, including hormones associated with insect development stages and the regulation of egg production in the ovaries of adult female insects, such as ecdysone, ecdysterone and juvenile hormones (JH) [51,52]. Sterol compounds derived from diet components, sesquiterpenes and peptide hormones at the initial stage of insect sex formation were also analyzed.

The methodology of sample preparation was simplified in order to minimize the number of steps and thus measure the uncertainty and preparation time. An important goal was to find such a method that the same extract could be used for several different chromatographic analyses, allowing the determination of these groups of compounds. Finally, a slightly modified procedure was adopted to separate disturbing fats, using their low solubility in acetonitrile. Thanks to this, an extract containing the compounds sought for a group of nonpolar compounds was obtained quite simply.

Sterol compounds were sought based on the derivation method with KOH proposed by Vanderplanck et al. [53]. The steps in sample preparation included homogenization of the material in methanol with 10% KOH (100 mg/1 mL) using a mortar, deprotonation with water and hexane, 10 min of extraction in an ultrasonic bath and purification of the substrate from the solid residue (fat) by centrifugation at 8000 rpm. The next step was a second extraction of the substrate with a mixture of dichloromethane–hexane (1:1 (2 × 1 mL)), followed by purification of the steroids on the SI column and flushing with diethyl ether. After evaporation of the ether, derivatization with N, O-bis (trimethylsilyl) trifluoroacetamide (BTSTFA) was performed to convert the fraction into higher volatility derivatives (i.e., trimethylsilyl esters (TMS)).

For the determination of JH and ecdysteroid substrates in one extraction, the method without a KOH derivate was used [54]. This method allowed the extraction of nonpolar compounds from a matrix containing fats, and in contrast to the KOH method, it allowed for the preservation of esters, which were later observed in the extract. It consisted of the homogenization of bark beetles in methanol (e.g., 1 g/3 mL) and extraction of nonpolar compounds (e.g., steroids, esters, ethers and fats) into hexane (2 × 3 mL), assisted by ultrasound. After centrifugation, the hexane with our compounds was taken and evaporated to dryness in a light stream of $N_2$. After evaporation, acetonitrile was added (100 μL for sample enrichment or an increase in concentration relative to the initial solution), in which the fats practically did not dissolve, and the solution was shaken at 50 °C.

Determination of the chemical compositions of the bark beetle samples was performed on a Clarus 600 gas chromatograph equipped with a PE-XLB chromatography column (30 m × 0.25 mm × 0.25 μm) and a Clarus 600 C mass detector. The chromatographic separation was carried out in the following conditions: dispenser temperature of 260 °C, initial temperature of the furnace of 130 °C (1 min) and a temperature rise of 12 °C/min to 300 °C. The registration of chromatograms in the SCAN mode covered a mass range from 100 to 620 m/z. Compounds were identified based on the analysis of spectrum of chromatographic peaks in the chromatograms, using the resources of the Wiley 8th Edition library. The samples intended for searching for steroid compounds were analyzed in SCAN mode, and the analyses were designed to detect the characteristic silyl ions of ecdysone and 20-hydroxyekdyson. The ions were selected after the SCAN analyses.

The effect of the sex and swarming period of insects on the compound concentration was examined using the GLM test. The differences in the mean values of the compound concentrations between swarming periods were analyzed. Since the variables were not normally distributed (proved by a Shapiro–Wilk normality test), the nonparametric Kruskal–Wallis test was applied. The statistical significance of the results was verified at a significance level of $\alpha = 0.05$. All of the statistical analyses were performed with Statistica 13 software.

## 3. Results and Discussion

Four known sterol compounds—β sitosterol, (3. β., 24R) ergost-5-en-3-yl, ergosta-5.7.22-trien-3 β-yloxy and cholecalciferol [55,56]—were detected in the adults of both sexes of *P. chalcographus* and *I. typographus* using the derivation method with KOH. Figure 2 shows an example of the retention times of these substances, with the spectra of compounds registered in SCAN mode allowing their identification.

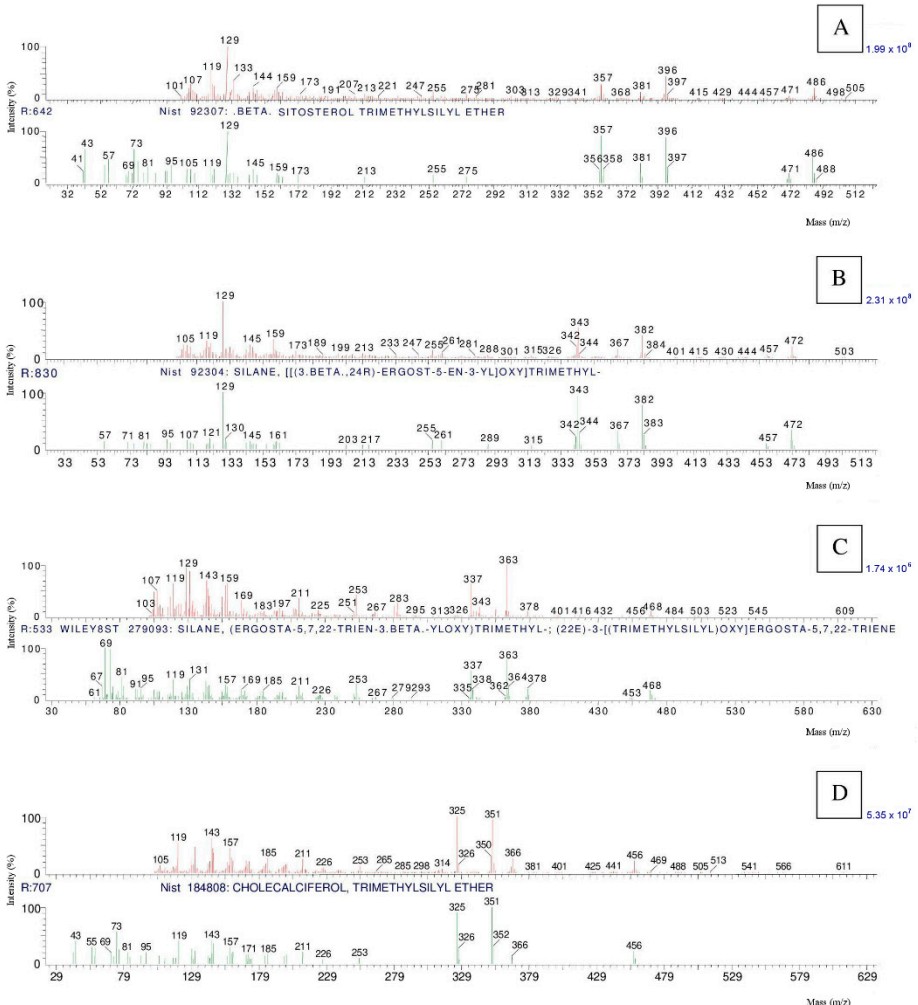

**Figure 2.** Chromatograms developed for the identification of sterol substances isolated from the adult bodies of *P. chalcographus* and *I. typographus* (method with KOH derivation) and the spectra of compounds ((**A**) β sitosterol, (**B**) (3. β.,24R) ergost-5-en-3-yl, (**C**) ergosta-5.7.22-trien-3 β-yloxy and (**D**) cholecalciferol). Red color = female, and green color = male.

Regardless of the insect species, the average concentrations of the analyzed substances reached the highest values in the case of cholecalciferol and the lowest for ergosta-5.7.22-trien-3 β-yloxy (Figure 3). Simultaneously, the largest difference in the average concentration of a substance between sexes was achieved by cholecalciferol (39.86% for *P. chalcographus* and 12.25% for *I. typographus*) and (3. β.,24R) ergost-5-en-3-yl (33.57% for *P. chalcographus* and 15.73% for *I. typographus*). All substances except the cholecalciferol extracted from the adult bodies of *P. chalcographus* showed higher compound concentrations in females than in males (Figure 3) but never reached a statistically significant level (Figure 3, Table 1), whereas the swarming period was a significant factor for the concentrations of the selected substances (Table 1). The average concentrations of β sitosterol and (3. β.,24R) ergost-5-en-3-yl in both sexes of *P. chalcographus* were significantly lower

in the second generation, with declines of 67.34% (Kruskal–Wallis test; $p = 0.0249$) and 63.61% (Kruskal–Wallis test; $p = 0.0502$), than the concentration of the substances isolated from the bodies of sister generation insects. In the case of *I. typographus* adults, the average concentration of ergosta-5.7.22-trien-3 β-yloxy per 1 g of specimens originating from the first generation reached higher values compared with the sister (98.8%, Kruskal–Wallis test; $p = 0.0305$) and second (93.22%, Kruskal–Wallis test; $p = 0.0384$) generations. For cholecalciferol, the mean value of the compound concentration in the second generation was 16.58 µg/g (N = 12), significantly higher than the first generation (mean concentration 7.27 µg/g; N = 12) (Kruskal–Wallis test; $p = 0.0035$).

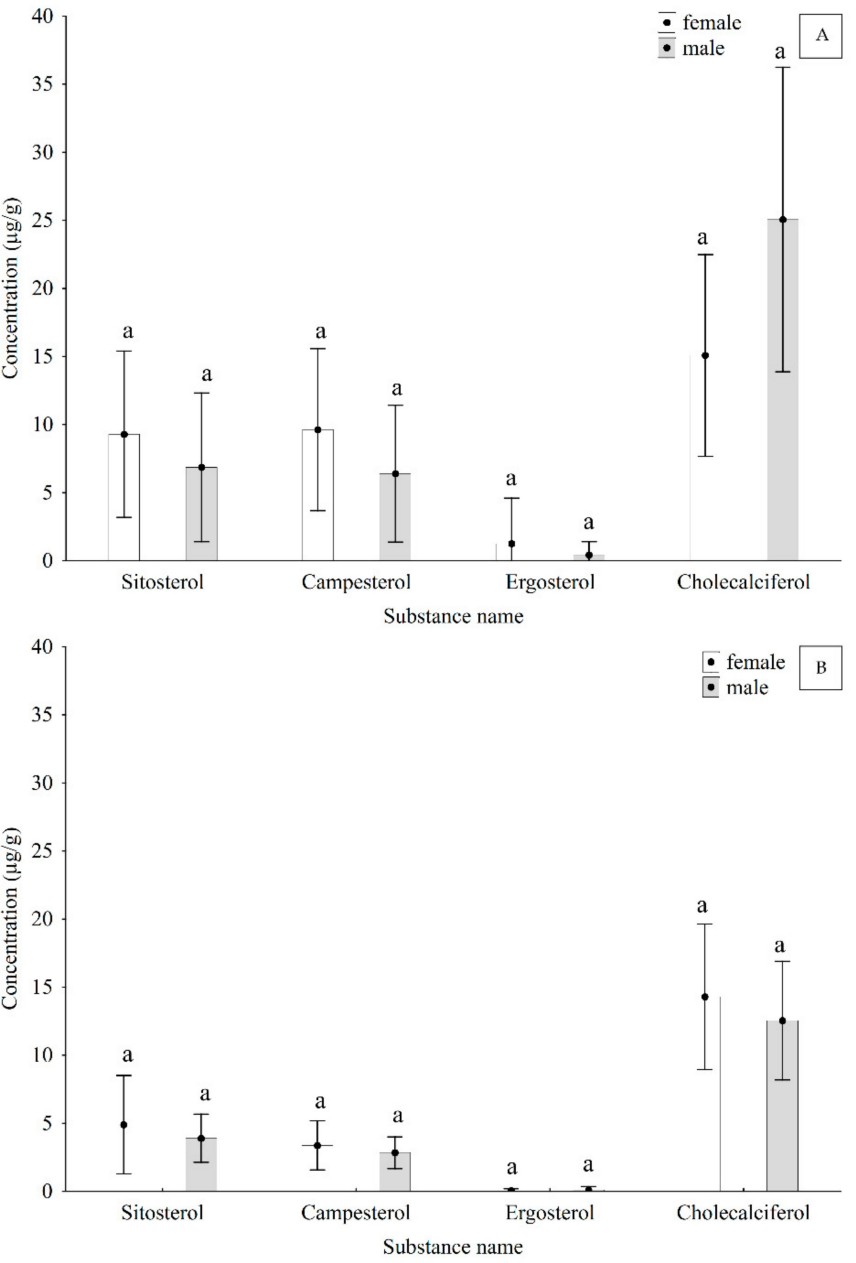

**Figure 3.** Mean (±SD) concentration of sterols (method with KOH derivation) differentiating the sexes of *I. typographus* (**A**) and *P. chalcographus* (**B**) isolated from adult specimens (the number of samples for each insect species was N = 12). The names of some substances are expressed as synonyms. Sitosterol = β sitosterol; campesterol = (3. β., 24R) ergost-5-en-3-yl; and ergosterol = ergosta-5.7.22-trien-3 β -yloxy. For each substances bars with the same letter do not differ significantly.

**Table 1.** Results of multivariate analysis of variance based on the general linear model (GLM) for the concentration of sterols, including the beetle sex and swarming period as quality variables.

| Effect | β sitosterol | | (3. β., 24R) ergost-5-en-3-yl | | ergosta-5.7.22-trien-3 β -yloxy | | Cholecalciferol | |
|---|---|---|---|---|---|---|---|---|
| | F | *p*-Value | F | *p*-Value | F | *p*-Value | F | *p*-Value |
| *P. chalcographus* | | | | | | | | |
| Sex | 2.027 | 0.1849 | 3.471 | 0.0920 | 0.971 | 0.3476 | 3.973 | 0.0742 |
| Swarming period | 4.959 | **0.0318** | 4.661 | **0.0371** | 2.755 | 0.1114 | 2.018 | 0.1835 |
| Sex and swarming period | 1.886 | 0.2018 | 2.092 | 0.1742 | 0.831 | 0.4635 | 0.141 | 0.8697 |
| *I. typographus* | | | | | | | | |
| Sex | 1.334 | 0.2632 | 0.791 | 0.3854 | 0.747 | 0.7006 | 0.599 | 0.4490 |
| Swarming period | 2.063 | 0.3565 | 2.400 | 0.1015 | 6.983 | **0.0026** | 7.908 | **0.0014** |
| Sex and swarming period | 2.199 | 0.1233 | 1.528 | 0.2414 | 0.479 | 0.3988 | 1.012 | 0.4104 |

*p*-value in bold indicates a statistically significant difference.

Based on the method without KOH derivation, in the obtained extracts of 10 sterols, 3 unidentified compounds and several dozen nonpolar compounds were detected (Table 2). From this group, after the removal of compounds of originating from plants, cholestenone, 4,6-cholestadien-3-on, choles-4en-3,6-dion and 17β-Hydroxyandrosta-1,4-dien-3-one benzoate were identified as potential steroid hormones, and in Figure 4, the retention times of these substances were 22.16, 22.97, 26.10 and 14.41 min, respectively, at a carrier gas flow of 1 cm$^3$ min$^{-1}$. The spectra of steroid compounds are presented in Figure S1.

The content of 17β-Hydroxyandrosta-1,4-dien-3-one benzoatein in the extracts from *P. chalcographus* was higher in the females than the males. A similar tendency was observed for the rest of the analyzed compounds and *I. typographus*; however, they reached lower levels (Figure 5).

Identification of the JH was impossible because of low concentrations and low intensities of peaks to separate the signal from the background. On the contrary, due to the quite distinctive molecular ions, a measurable signal for ecdysone and ecdysterone was obtained. Low concentrations of both steroid hormones were observed in the cases of both bark beetle species. The concentration of 20-hydroxyecdysone (20E) differed in the analyzed bark beetles. The higher amount was observed in the male bodies of *P. chalcographus*, whereas in the case of the *I. typographus* species, the relationship was inverted. However, in the case of ecdysone, the results showed a slightly higher concentration in favor of males in both bark beetle species (Figure 6). This result agrees partly with De Loof [4], who indicated the possibility of females using the substance produced by the ovaries during egg laying, while ecdysone can act as a sex hormone in males. Zhang et al. [57] proved that ecdysone participates in the production of male-type neurons. Nevertheless, neither the analyzed substances nor the bark beetle species differences in concentrations between sexes were statistically significant. The lack of clear differences in the contents of ecdysteroids in the bodies of female and male bark beetles may be related to the developmental stage of the insects. It is suspected that the highest levels of ecdysteroids and the resulting differences between sexes may be found in the pupae stages, when the reproductive behavior of holometabolous insects develops [52,58,59]. The compounds found in the bark beetle adults could therefore only be the remains of metamorphosis, hence the lack of differences in concentrations between the bodies of males and females.

**Table 2.** List of sex-differentiating substances isolated from the adult bodies of *I. typographus* and *P. chalcographus*, originating from Theysohn pheromone traps and specifying their function * (PH = probable pheromone, S = sterol, H = probable hormone, O = indifferent (i.e., neither pheromone, either hormone), T = terpene and UI = unidentified). Substances that are potential steroid hormones are marked in bold.

| Compound | RT | Ion Molecular | Cas No. | Function * |
|---|---|---|---|---|
| vitamin E | 19.138 | 430.00 | 59-02-9 | PH |
| cholesterol | 19.850 | 386.00 | 57-88-5 | S |
| provitamin d3 | 20.573 | 384.00 | 434-16-2 | PH |
| campesterol | 21.550 | 400.00 | 474-62-4 | S |
| **cholestenone** | **22.165** | **384.12** | **601-54-7** | **SH** |
| **4,6-cholestadien-3-one** | **22.977** | **382.14** | **566-93-8** | **SH** |
| gamma sitosterol | 18.843 | 414.00 | 83-47-6 | S |
| alpha n-hexadecylhydrindane | 23.933 | 384.96 | 55401-73-5 | PH |
| ui not sterol | 24.146 | 321.24 | | PH |
| ui sterol | 25.875 | 316.00 | | S |
| **cholest-4-ene-3,6-dione** | **26.105** | **398.14** | **984-84-9** | **SH** |
| ui rather sterol | 29.121 | 347.26 | | S |
| ui rather sterol | 29.750 | 349.26 | | S |
| methyl hexadecanoic acid | 9.479 | 74.00 | 112-39-0 | O |
| methyl-6,9-octadecadienoate | 10.898 | 294.00 | 56599-55-4 | PH |
| methyl 7,10,13-hexdecatrienate | 10.98 | 79.00 | 13058-55-4 | PH |
| methyloctadecanoic acid | 11.09 | 74.00 | 112-61-8 | O |
| unsaturated acid methyl ester | 11.481 | 81.00 | | PH |
| 2-octadecanal (synthetic pheromone) | 11.459 | 83.00 | 56554-96-2 | PH |
| ui silacyclobutane derivative | 11.805 | 72.00 | | PH |
| n-propyl-9-cis,11-trans-octadecanoate | 11.463 | 81.26 | 336-72-6 | PH |
| n-propyl-9-octadecadienoate- | 11.470 | 265.00 | 336-64-8 | PH |
| ethyl 9,12,15-octadecatrienoate | 11.467 | 79.00 | 336-77-4 | PH |
| tricosane (most likely) | 12.269 | 85.00 | 638-67-5 | O |
| methyl 10-oxooctadecanoate | 12.592 | 156.00 | 870-10-0 | PH |
| ui silacyclobutane derivative (most likely) | 13.110 | 72.00 | | PH |
| 1-docosanol | 13.243 | 83.00 | 661-19-8 | PH |
| ui alcohol | 13.559 | 83.00 | | PH |
| ui alcohol | 13.559 | 83.00 | | PH |
| ui saturated hydrocarbon | 13.656 | 71.00 | | O |
| phtalic acid 6-ethyloct-3-yl-2-ethylhexyl ester | 14.138 | 149.00 | 900315-53-8 | PH |
| **17β-hydroxyandrosta-1,4-dien-3-one benzoate** | **14.411** | **122.00** | **19041-66-8** | **SH** |
| ui alcohol | 14.832 | 83.00 | | PH |
| methyl 11,14,17-eicosatrienoate | 14.893 | 108.00 | 90041-66-8 | PH |
| ui saturated hydrocarbon | 14.832 | 71.00 | | O |
| ui aromatic hydrocarbon | 15.044 | 119.00 | | O |
| 5-oxazolidinone, 3-benzoyl-2-(1,1-dimethylethyl)-4-phenylmethyl- | 15.590 | 105.00 | 104057-67-2 | PH |
| 2,6,10,14,18,22-tetracosahexaene, 2,6,10,15,19,23-hexamethyl-, (squalene) | 15.748 | 81.00 | 7683-64-9 | PHT |
| ui cyclic hydrocarbon | 16.040 | 82.00 | | PH |
| ui cyclic hydrocarbon | 16.032 | 82.00 | | PH |
| methyl 2-hydroxy-6-hexadeca-8,11,14-trienyl benzoic acid | 16.036 | 108.00 | | PH |
| ui aromatic hydrocarbon | 16.320 | 119.00 | | O |
| ui aromatic hydrocarbon | 16.320 | 119.00 | | O |
| 2-methyl-2-phenyl pentadecane | 16.403 | 105.00 | | PH |
| trans squalene (most likely) sterol precursor | 16.733 | 137.00 | | PHT |

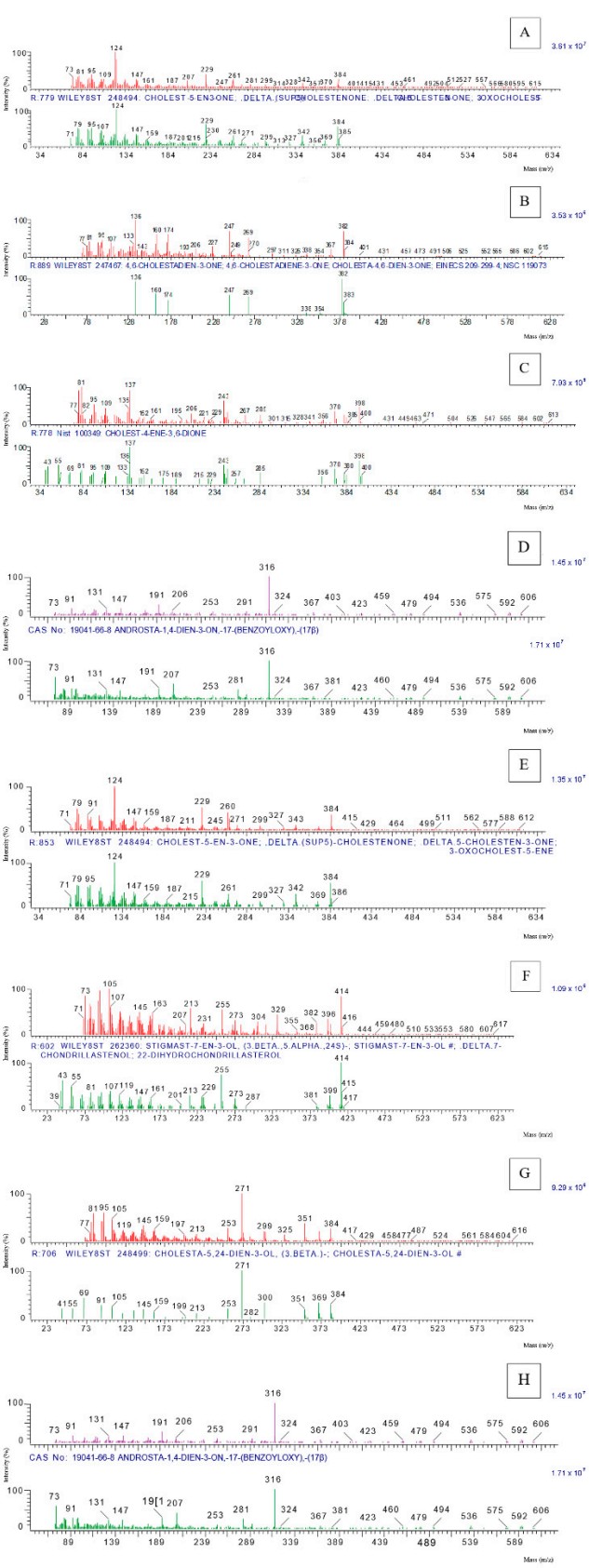

**Figure 4.** Chromatograms for the identification of sterol substances that could potentially act as sex hormones, isolated from the adult bodies of *P. chalcographus* (method without KOH derivation) (**A–D**) and *I. typographus* (**E–H**) species. Red color = female, and green color = male.

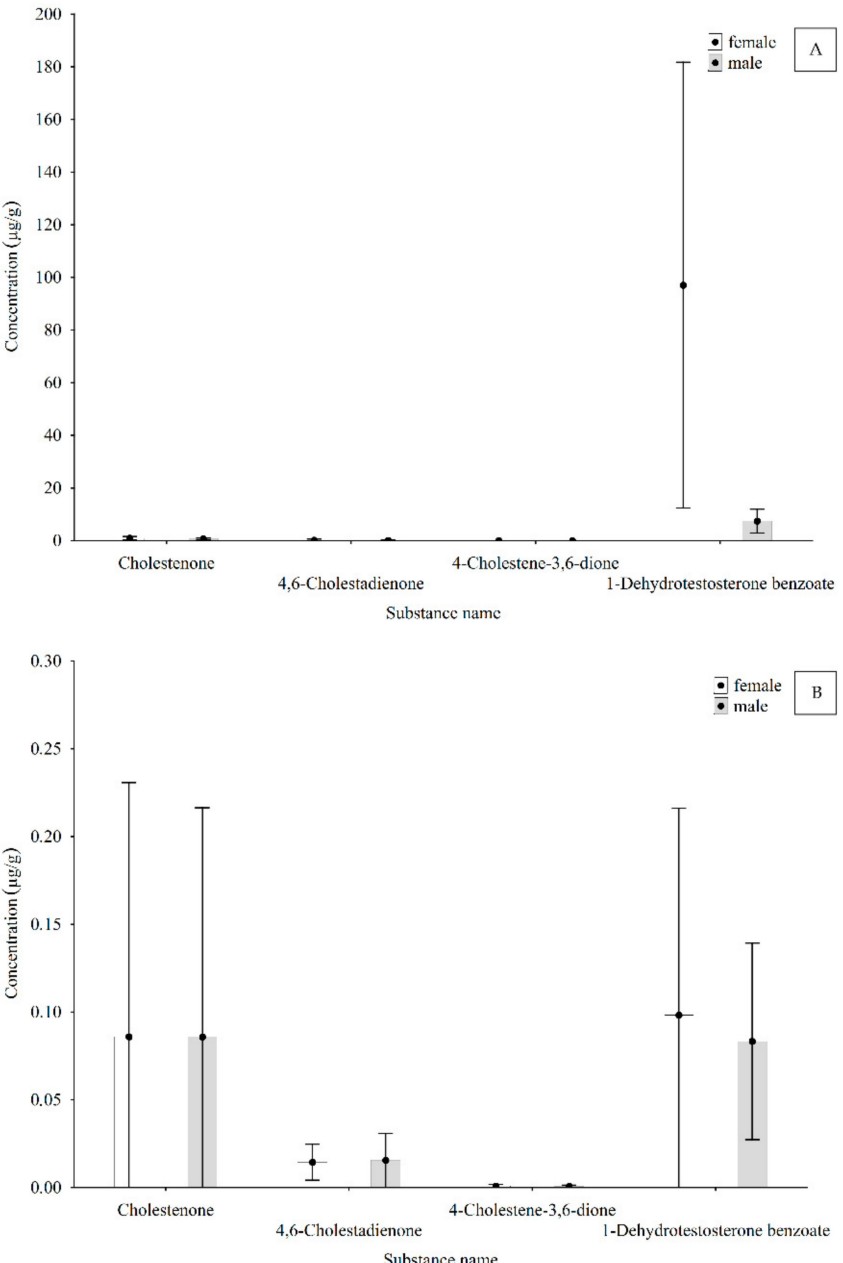

**Figure 5.** Mean (±SD) concentration of sterol compounds (method without KOH derivation) differentiating the sexes of *I. typographus* (**A**) and *P. chalcographus* (**B**) isolated from adult specimens (the number of samples for each insect species was N = 12). The names of some substances have been expressed as synonyms. 1-Dehydrotestosterone benzoate = 17β-Hydroxyandrosta-1,4-dien-3-one benzoate.

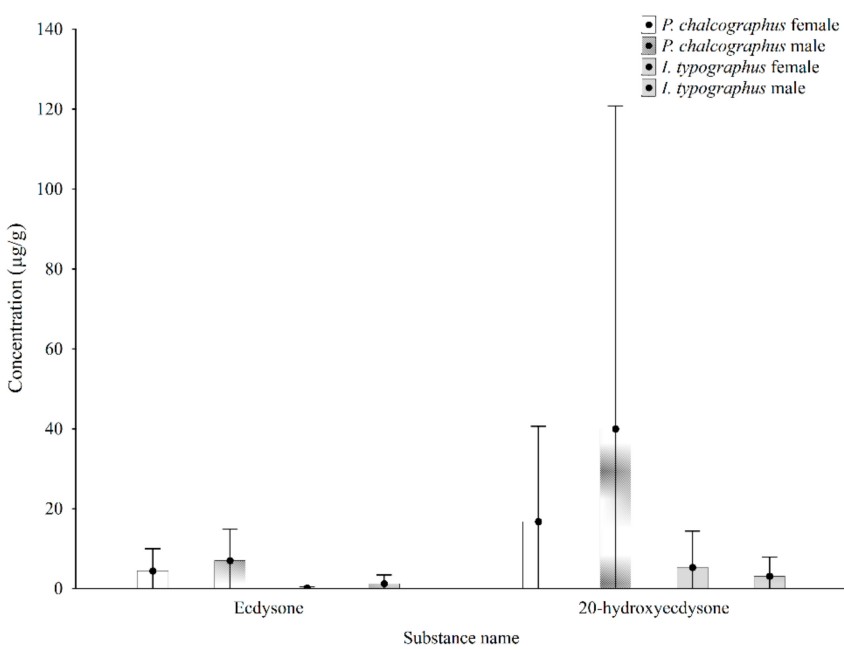

**Figure 6.** Mean (±SD) concentration of ecdysone and 20-hydroxyecdysone in the bodies of the most dangerous Norway spruce bark beetle species (the number of samples for each insect species was N = 12).

## 4. Conclusions

Based on the presented research, it was found that in the bodies of the studied bark beetle adults, the presence of ecdysteroids (i.e., ecdysone and 20-hydroxyekdysone) was confirmed. In addition, four compounds were isolated that were identified as cholesterol derivatives (that could potentially act as hormones). Compounds from the ecdysteroid group found in bark beetle bodies should be compared with the standards to avoid misidentification (difficulties in detecting compounds occurring in the shade of substances with similar chemical structures). Due to the lack of appropriateness or high cost of the standards, it is only possible to assume the identification of both compounds. If further work to determine the possibility of assessing the share of males and females in the mass of specimens taken from the traps is conducted, it would be necessary to focus on measuring the content of these compounds, but based on standards, especially in the case of ecdysones, the compounds of JH, 17β-Hydroxyandrosta-1,4-dien-3-one benzoate and sterols, whose concentrations in the extracts were very low, made it difficult to assess the differences in male and female tissues. Due to the unique nature of the research and its importance for forest protection, as the gender structure of bark beetles is key to distinguishing the outbreak phase, these studies should be continued.

**Supplementary Materials:** The following are available online at https://www.mdpi.com/article/10.3390/f12050536/s1. Figure S1: Spectra of potential steroid hormones (A- cholestenone, B- 4,6-cholestadien-3-on, C- choles-4en-3,6-dion, D- 17β-Hydroxyandrosta-1,4-dien-3-one benzoate) isolated from adult bodies of *I. typographus* and *P. chalcographus*.

**Author Contributions:** Conceptualization, M.K. and B.B.; methodology, M.K., B.B. and M.C.; formal analysis, M.C. and M.K.; writing—original draft preparation, M.K. and B.B.; writing—review and editing, M.K., B.B. and M.C.; visualization, M.K. and B.B. All authors have read and agreed to the published version of the manuscript.

**Funding:** This research was funded by the Ministry of Science and Higher Education, Poland (SUB/040013-D019), and the University of Agriculture in Krakow - research grant number 4427/2013.

**Data Availability Statement:** The data presented in this study are available on request from the corresponding author. The data are not publicly available due to policy of the institute.

**Acknowledgments:** The authors would like to thank anonymous reviewers for the thorough assessment of this paper and for many valuable and helpful suggestions.

**Conflicts of Interest:** The authors declare no conflict of interest. The funders had no role in the design of the study; in the collection, analyses or interpretation of the data; in the writing of the manuscript; or in the decision to publish the results.

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
