# Peer review of "Attempt to Identify Sex Hormones in the Bodies of Selected Norway Spruce Bark Beetles"

_forests, doi:10.3390/f12050536_

Round 1
Reviewer 1 Report
Authors tried to distinguish sex hormones of N. spruce bark beetles. Due to outbrake of bark beetle and decline of European forest due to infestation casused by draught, it is extremely important to undestand the hormones of bark beetles.
The english is written in perfect way, the text is easy to read and follow and attracts the reader to continue reading.
There are just minor thinks to be checked:
Title: Shouldn´t the title rather be: "Attempt to identify sex hormones in the bodies of selected Norway spruce bark beetles"?
Introduction: It would be good to add information about 2-methyl-3-buten-2-ol as I. typographus hormone occuring also in gas phase, which is, however, at the same time produces by the N. spruce trees itself. Moreover, there is a study, which was done at Beskydy mountains too. See 10.1016/j.agrformet.2016.10.005 for details.
line 166: N2 should be small 2; use micro (µ) sign instead of u
line 173: should be small t in temperature
Figure 1: having the picture on the back with shadings of the graph is very unusual. Please remove the picture. It is not clear for which specimen refers the y axes, even it is described in the text bellow. It would be good to describe it even in the graph, as follows (for example): Number of trapped individuals I. typographus and yy axis as Number of trapped individuals P. chalcographus. Moreover, write Ips as I. and this be consistent with writing P. chalcographus
lines 191-192: why you do not write β instead of beta?
Figure 2 and 4: The time should have unit
line 202: should be big R in Red. Use β
Figure 3, 5 and 6: Description of y axis should be: Concentration [µg/g]. It would be nicer too to have the font size of both the A and B the same.
Table 1: Check the decimal points to be everywhere as dots.
line 230: should be "of 10 sterols"
lines 232-234 and 237: why are you using capitals?
line 235: "respectively" should be at the end of the sentence
Table 2: use decimals as dots in ion molecular row. Why there are ? in compound names? What does it mean ME in OCTADECANOIC ACID, HEXADECANOIC ACID, ESTER ME KWASU NIENASYCONEGO, 2-HYDROXY-6-HEXADECA-8,11,14-TRIENYL BENZOIC ACID ME? Check the names - METHYL 6,9-OCTADECADIENOATE should be methyl-6,9-octadecadienoate, the same for METHYL 7,10,13-HEXDECATRIENOATE and ETHYL 9,12,15-OCTADECATRIENOATE, METHYL 11,14,17-EICOSATRIENOATE. Correct the name of 5-OXAZOLIDINONE,-3-BENZOYL-2-(1,1-DIMETHYLETHYL),-4(PHENYLMETHYL) and 2,6,10,11,18,22-TETRACOSAHEXAENE,-2,6,10,15,19,23-HEXAMETHYL (SQUALEN). There are commas, which should not be there. All the chemicals should be written together (e.g. it should be 2-methyl-2-phenylpentadecane etc.), whoch is correct in the graphs. Why do you use capitals? I think the names should be written normally.
line 244-245: should be together with Table description above the Table
Figure 5: check the corectness of the chemicals, for example "androsta-1,4-dien-3-on17(benzoyloxy),(17beta)" should be 17β-(benzoyloxy)androsta-1,4-dien-3-one to be consistent with the semiofficial naming used in the manuscript.
Figure 6: instead of 20-OH write 20-hydroxyecdysone
line 288: write "four" instead of 4. Numbers lower than 10 are usually written as text (if not being decimals).
line 296-297: do not use capitals
line 299: Sentence "The bigger samples need to be analyzed in order to become visible in the chromatograms" should stay alone.
Conclusions: Almost half of the conclusion is the outlook to the future, what should be done next - that is too long. Please, conclude longer what you have done and the outlook should be shorter.
Author Response
Authors tried to distinguish sex hormones of N. spruce bark beetles. Due to outbrake of bark beetle and decline of European forest due to infestation casused by draught, it is extremely important to undestand the hormones of bark beetles.
The english is written in perfect way, the text is easy to read and follow and attracts the reader to continue reading.
There are just minor thinks to be checked:
Title: Shouldn´t the title rather be: "Attempt to identify sex hormones in the bodies of selected Norway spruce bark beetles"?
Thank you for your suggestion. We have changed the title to the version that was proposed.
Introduction: It would be good to add information about 2-methyl-3-buten-2-ol as I. typographus hormone occuring also in gas phase, which is, however, at the same time produces by the N. spruce trees itself. Moreover, there is a study, which was done at Beskydy mountains too. See 10.1016/j.agrformet.2016.10.005 for details.
Thank you for your suggestion. We added the information about 2-methyl-3-buten-2-ol, produced both by insects and host trees in the proper place of Introduction chapter (lines: 38-40)
line 166: N2 should be small 2; use micro (µ) sign instead of u
We have made corrections according to your suggestion.
line 173: should be small t in temperature
Thank you for pointing out the shortcoming. The error in the text have been corrected.
Figure 1: having the picture on the back with shadings of the graph is very unusual. Please remove the picture. It is not clear for which specimen refers the y axes, even it is described in the text bellow. It would be good to describe it even in the graph, as follows (for example): Number of trapped individuals I. typographus and yy axis as Number of trapped individuals P. chalcographus. Moreover, write Ips as I. and this be consistent with writing P. chalcographus
Thank you for your comment. According to your suggestion we have corrected the figure 1 to make it more clearer. We removed the photo from the background of the figure, changed the descriptions of the y1 and y2 axes and corrected I. typographus name in the legend.
lines 191-192: why you do not write β instead of beta?
Thank you for your comment. The names of the substances quoted in the text are taken from the CAS database. Based on chemical nomenclature, both chemical compounds names, i.e. beta and β are correct. Nevertheless, according to your suggestion we have changed "beta" on"β" through the whole manuscript.
Figure 2 and 4: The time should have unit
According to your suggestion we have added unit "minute" [min] for retention time.
line 202: should be big R in Red. Use β
We have made corrections according to your suggestion.
Figure 3, 5 and 6: Description of y axis should be: Concentration [µg/g]. It would be nicer too to have the font size of both the A and B the same.
Thank for your comment. We have made correction in the figures according to your both suggestions.
Table 1: Check the decimal points to be everywhere as dots.
Thank for your comment. We made sure that there were dots everywhere when entering values after the decimal point.
line 230: should be "of 10 sterols"
We have made correction in the text according to your suggestion.
lines 232-234 and 237: why are you using capitals?
Thank you for your comment. The names of the chemicals in the text can be written with either capitals or lower case letters. In order to standardize the spelling of names, we adopted uncapitalized letters.
line 235: "respectively" should be at the end of the sentence
We have made correction in the text according to your suggestion.
Table 2: use decimals as dots in ion molecular row.
We have made correction in the text according to your suggestion.
Why there are ? in compound names? What does it mean ME in OCTADECANOIC ACID, HEXADECANOIC ACID, ESTER ME KWASU NIENASYCONEGO, 2-HYDROXY-6-HEXADECA-8,11,14-TRIENYL BENZOIC ACID ME? Check the names - METHYL 6,9-OCTADECADIENOATE should be methyl-6,9-octadecadienoate, the same for METHYL 7,10,13-HEXDECATRIENOATE and ETHYL 9,12,15-OCTADECATRIENOATE, METHYL 11,14,17-EICOSATRIENOATE. Correct the name of 5-OXAZOLIDINONE,-3-BENZOYL-2-(1,1-DIMETHYLETHYL),-4(PHENYLMETHYL) and 2,6,10,11,18,22-TETRACOSAHEXAENE,-2,6,10,15,19,23-HEXAMETHYL (SQUALEN). There are commas, which should not be there. All the chemicals should be written together (e.g. it should be 2-methyl-2-phenylpentadecane etc.), whoch is correct in the graphs. Why do you use capitals? I think the names should be written normally.
Thank you for pointing out some inaccuracies in substance nomenclature. The question mark used together with some compounds names indicated the most likely assignment of chromatograms to the specific substances. In order to dispel any doubts, the question mark has been replaced with "the most likely" term "ME" is the abbreviation for methyl group. All suggestions (corrected names, uncapitalized letters were included in the text).
line 244-245: should be together with Table description above the Table
We agree with reviewer suggestion and moved this part to the table 1 description above the Table.
Figure 5: check the corectness of the chemicals, for example "androsta-1,4-dien-3-on17(benzoyloxy),(17beta)" should be 17β-(benzoyloxy)androsta-1,4-dien-3-one to be consistent with the semi official naming used in the manuscript.
Thank you for your comment. We have made correction in the text by adopting a single name for the analyzed substances throughout the manuscript. In order to improve the readability of the x-axis in figures 3 and 5, we used abbreviations or synonyms in the graphs and added the full name of the molecules in the captions.
Figure 6: instead of 20-OH write 20-hydroxyecdysone
We have made correction in the text according to your suggestion.
line 288: write "four" instead of 4. Numbers lower than 10 are usually written as text (if not being decimals).
Thank you for your comment. We have made correction in the text according to your suggestion.
line 296-297: do not use capitals
We have made correction in the text according to your suggestion.
line 299: Sentence "The bigger samples need to be analyzed in order to become visible in the chromatograms" should stay alone.
Thank you for your comment. According to your next suggestion we have shorten "Conclusions" chapter and removed this sentence.
Conclusions: Almost half of the conclusion is the outlook to the future, what should be done next - that is too long. Please, conclude longer what you have done and the outlook should be shorter.
We have improved "Conclusions" according to your suggestions.

Reviewer 2 Report
Review of MS 2021, 12, “Attempt to identify of sex hormones in the bodies of selected Norway spruce bark beetles
The MS faces an interesting study concerning two important bark beetles such as Ips typograhus (L.) and Pityogenes chalcographus (L.) in Norway spruce. The two species are in fact known for the damage caused in various conifers and in particular on Norway spruce. Overall, the work contributes to improving information on the chemical compounds of the species related to possible interactions between the sexes and the environment. The results presented in the MS could, albeit partially, contribute to forest protection including improving the critical role of monitoring for these two species.
Overall, however, the introductory part must be simplified as it is very complex and often not always clear. Continued references to the vertebrate hormone system should be limited. The authors should better highlight in the text the importance of improving the procedures for determining sex, often not simply detectable through morphological characters.
Different clarifications and simplifications are requested from the authors
Line 27-28 Add reference for this sentence;
Line 35-38. Considering the numerous and important considerations, it is advisable to insert bibliographic references;
Line 74-82. Simplify this paragraph, as it is too complex;
Line 116. Authors can specify how often pheromone lures were changed;
Line 119. The authors do not believe that the beetle collection date every 8 days is an excessive period as individuals in traps often die within a few days. This effect may have influenced the analyses.
Line 120-121. I wonder why the authors choosed beetles for chromatrographic analyses only those of 18th May and 1st September. The authors did not believe they were also carrying out the analyses for the catches of the generation that emerges between 7 and 14 July;
Line 122-123. Which morphological characters were used for the determination of the sexes of the two species?
Line 150-153. How many beetles were used for each sample and in relation to the analyses were processed in the analyses;
Line 182-183, Kruskal-Wallis test is a non-parametric method to verify the equality of the medians of different groups. Specify why this method is used. Was the sampled number of beetles low ?
Line 186. The graph shows that the counts and therefore the monitoring of beetles occur every seven days and not every eight days as written in the materials and methods.
Line 237-239. I was wondering if it is necessary to use capital letters for molecules and the same for table 2
In order to improve the readability of the x-axis in figure 5, it is possible to use abbreviations or synonyms in the graph (see PubChem) and add the full name of the molecules in the caption;
Figure 3. 5 and Figure 6 show the average with the SD but for each average it is advisable to specify the number of samples.
Author Response
Dear Reviewer,
Thank you very much for your constructive remarks and suggestions helping to improve our manuscript. We agree with the all of them. Below please find our detailed answers, whereas all changes you can find in the body of attached files.
Sincerely
Authors
The MS faces an interesting study concerning two important bark beetles such as Ips typograhus (L.) and Pityogenes chalcographus (L.) in Norway spruce. The two species are in fact known for the damage caused in various conifers and in particular on Norway spruce. Overall, the work contributes to improving information on the chemical compounds of the species related to possible interactions between the sexes and the environment. The results presented in the MS could, albeit partially, contribute to forest protection including improving the critical role of monitoring for these two species.
Overall, however, the introductory part must be simplified as it is very complex and often not always clear. Continued references to the vertebrate hormone system should be limited. The authors should better highlight in the text the importance of improving the procedures for determining sex, often not simply detectable through morphological characters.
Thank you for your comment. We have improved introduction chapter according to the reviewer suggestions.
Different clarifications and simplifications are requested from the authors
Line 27-28 Add reference for this sentence;
According to your suggestion we have added relevant references.
Line 35-38. Considering the numerous and important considerations, it is advisable to insert bibliographic references;
Thank you for your comment. According to your prior suggestion, we have simplified the introduction chapter therefore the section at lines 35-38 (in revised manuscript lines: 42-45) was also removed
Line 74-82. Simplify this paragraph, as it is too complex;
According to your comment, we have rewritten this paragraph.
Line 116. Authors can specify how often pheromone lures were changed;
Thank you for your comment. Both pheromone lures dedicated to analyzed bark beetles species were changed once during the control period, i. e. on July 15. This information was added in the text.
Line 119. The authors do not believe that the beetle collection date every 8 days is an excessive period as individuals in traps often die within a few days. This effect may have influenced the analyses.
Thank you for your comment. During the insects sampling from pheromone traps we had collected only alive specimens.
Line 120-121. I wonder why the authors choosed beetles for chromatrographic analyses only those of 18th May and 1st September. The authors did not believe they were also carrying out the analyses for the catches of the generation that emerges between 7 and 14 July;
Thank you for your comment. The insects were taken from several inspection dates not only on May 18 and September 1. For each insects generation samples from 3 terms have been analyzed. Our results present the values ​​for the generation, not for these terms. Data from several terms were analyzed together when they related to a given generation. It has been clarified in the text.
Line 122-123. Which morphological characters were used for the determination of the sexes of the two species?
The procedure used for the determination of the sexes of two bark beetles species were explained in detail in the text (lines 149-152): “Gender identification of P. chalcographus adults was based on the size of the teeth on elytral humerus, which is bigger in males in contrast to females, much smaller [55]). In case of I. typographus specimens assigning to sex was based on the genital organs extraction.”
Line 150-153. How many beetles were used for each sample and in relation to the analyses were processed in the analyses;
Thank you for your comment. The number of bark beetles analyzed corresponded to the total amount of 10 g per each swarming period and sex and varied, depending on bark beetles species and the sex, since the fresh weight of species differ significantly - females are heavier than males in respect to the both I. typographus and P. chalcographus. It was successfully proved in our prior research (Bednarz, B.; Kacprzyk, M. 2012. An innovative method for sex determination of the European spruce bark beetle Ips typographus (Coleoptera: Scolytinae). Entomol. Gen. 2012, 34 (1-2), 111-118.) Therefore the unit we refer was weight in grams, which corresponds to about 150 specimens of I. typographus and 2000 specimens of P. chalcographus per 1 g. Information on the number of insects included in the samples have been explained in the chapter "Materials and methods".
Line 182-183, Kruskal-Wallis test is a non-parametric method to verify the equality of the medians of different groups. Specify why this method is used. Was the sampled number of beetles low ?
Thank you for your comment. The Kruskal-Wallis test was chosen prior to normality test of variables. Since the variables had been non-normal distributed (proved by Shapiro-Wilk normality test) the nonparametric test was applied. Number of samples analyzed and information of Shapiro-Wilk normality test have been added in both text of manuscript in Chapter "Materials and methods”- Lines: 200-202.
Line 186. The graph shows that the counts and therefore the monitoring of beetles occur every seven days and not every eight days as written in the materials and methods.
Thank you for your comment. An inaccuracy had crept into the text. The monitoring of beetles occured, as it was expressed in the figure 1, every seven days. It has been changed already in the manuscript text.
Line 237-239. I was wondering if it is necessary to use capital letters for molecules and the same for table 2
In order to improve the readability of the x-axis in figure 5, it is possible to use abbreviations or synonyms in the graph (see PubChem) and add the full name of the molecules in the caption;
Thank you for your comment. We had changed the capital letters for molecules in both table 2, figures 3 and 5 follow your suggestion. At the same time instead of using long names we used synonyms of the substances names in the mentioned figures. According to your suggestion the full name of the molecules were mentioned in the figures caption.
Figure 3. 5 and Figure 6 show the average with the SD but for each average it is advisable to specify the number of samples.
Thank you very much for your comment. The number of samples was added in the captions of figures.

Round 2
Reviewer 2 Report
Dear authors,
The changes requested by me have been made in the text. In particular, the various simplifications are now more evident. In this phase, you are asked to pay close attention to the name of the indicated molecules by uniforming the whole.